# Treatment Harms in Paediatric Primary Care

**DOI:** 10.3390/ijerph20146378

**Published:** 2023-07-17

**Authors:** David M. Reith, Sharon Leitch, Kyle Eggleton, Katharine Wallis, Steven Lillis, Martyn Williamson, Wayne Cunningham

**Affiliations:** 1Dean’s Department, Otago Medical School, University of Otago, Dunedin 9016, New Zealand; 2Department of General Practice and Rural Health, Dunedin School of Medicine, University of Otago, Dunedin 9016, New Zealand; sharon.leitch@otago.ac.nz (S.L.); martyn.williamson@otago.ac.nz (M.W.); 3Faculty of Medical and Health Sciences, University of Auckland, Auckland 1010, New Zealand; k.eggleton@auckland.ac.nz; 4Mayne Academy of General Practice, Medical School, The University of Queensland, Brisbane 4067, Australia; k.wallis@uq.edu.au; 5Department of General Practice and Primary Health Care, Faculty of Medical and Health Sciences, University of Auckland, Auckland 1010, New Zealand; steven.lillis@outlook.co.nz; 6Independent Researcher, Upper Hutt 5018, New Zealand

**Keywords:** drug utilisation, paediatric, adolescent, harm, general practice

## Abstract

The aim of this study was to describe the epidemiology in children of harms detectable from general practice records, and to identify risk factors. The SHARP study examined 9076 patient records from 44 general practices in New Zealand, with an enrolled population of 210,559 patients. “Harm” was defined as disease, injury, disability, suffering, and death, arising from the health system. The age group studied was ≤20 years of age. There were 193 harms to 141 children and adolescents during the 3-year study period. Harms were reported in one (3.5%) patient aged <2 years, 80 (6.6%) aged 2 to <12 years, 36 (4.9%) aged 12 to <18 years, and 24 (7.5%) aged 18 to ≤20 years. The annualised rates of harm were 36/1000 child and adolescent population for all harms, 20/1000 for medication-related harm (MRH), 2/1000 for severe MRH, and 0.4/1000 for hospitalisation. For MRH, the drug groups most frequently involved were anti-infectives (51.9%), genitourinary (15.4%), dermatologicals (12.5%), and the nervous system (9.6%). Treatment-related harm in children was less common than in a corresponding adult population. MRH was the most common type of harm and was related to the most common treatments used. The risk of harm increased with the number of consultations.

## 1. Background

Children are a vulnerable patient group and, therefore, may be at increased risk of harm during medical treatment. Harm is different to safety incidents, adverse drug events (ADE), and medication error because these other events may, or may not cause harm, and harm can occur without there being error. We define “harm” as a disease, injury, disability, suffering, or death, arising from the health system. There is a paucity of information about the extent of the risk of harm in primary care because research in this field is limited. More is known about these other types of event and in adult populations [1].

In a New Zealand inpatient paediatric population, previous research found the rate of adverse drug events (ADEs) was 2.1 per 100 prescription episodes, 12.9 per 100 admissions, and 22.1 per 1000 patient days [2]. Forty-six percent of the ADEs were classified as being serious, 15% were deemed to result in persistent disability or were classified as life threatening. However, the rates of ADEs in children treated in primary care are unknown and there may be considerable under-reporting. A UK study found that of all incidents reported in children in the UK, only 4% were reported from primary care, a figure that appears implausibly low [3].

The true rate of medication error is unknown as the data have come from spontaneous reporting, a method that is known to underestimate error rates [4]. The most commonly reported types of incidents in children in the UK are medication incidents (17%), treatment/procedure incidents (13%), and patient accidents (11%) [3]. Around 50% of medication errors with vaccines involve administration of the wrong vaccine [4].

Factors that might predispose children to harm include the complexity of dosing medicines, differing pharmacokinetics to adults, a lack of suitable formulations for children, and errors in administration. Dose error may be contributed to by a lack of dosing information for children, leading to off-label prescribing [4]. Drug handling (pharmacokinetics) and drug action (pharmacodynamics) varies between different age groups. Hence, the age of the patient, and the medication involved, affect the risk of overdose or underdose [4].

## 2. Aims

The aim of this study was to describe the epidemiology in children of harms detectable from general practice records, and to identify risk factors. 

## 3. Methods

The SHARP study is a stratified, 2-level cluster, retrospective records review study [5]. Clusters were based on the practice size and rurality. Practice size was determined by the number of enrolled patients, divided into tertiles, creating three groups: small-, medium-, and large-sized practices. Rurality was defined by Statistics NZ (2004 classification), apart from practices which were based in “independent urban communities”, which were included in the rural group due to a lack of secondary care support and the rural location of patients who attend those practices. The study examined 9076 patient records from 44 general practices in New Zealand, with an enrolled population of 210,559 patients. There were 24 rural practices and 20 urban practices. The characteristics of the practices were representative of New Zealand general practices [6]. Patients were selected at random from those enrolled with the general practices chosen for inclusion in the study, and the sampling was balanced by practice size and location. The records for each selected patient were reviewed for a 3-year period, 2011 to 2013 inclusive.

This current study is a sub-group analysis of the SHARP study. The age group for the purposes of this study was from birth to ≤20 years of age, consisting of 2300 SHARP patients. Males and females were included and there were no exclusion criteria. Age subgroups were based on the age groups used for medicines regulation by the FDA, namely birth to <2 years, 2 years to <12 years, and 12 years to <18 years. Data from the 18 to ≤20-year age group is also included as these overlap with census population strata, which in New Zealand is reported in 5-year age bands. Social deprivation was determined using the geographically based NZDep Index [7].

“Harm” was defined as disease, injury, disability, suffering, and death, arising from the health system. Harm was identified by the investigators from a longitudinal review of the selected individual patient records. Categorisation of patient harm (including type, preventability, severity, and outcome) was coded using the Medical Dictionary for Regulatory Activities (MedDRA) 18.0. 

Preventability was coded using the definitions developed by McKay et al. 2013, as “not preventable and originated in secondary care”, “preventable and originated in secondary care OR not preventable and originated in primary care”, “potentially preventable and originated in primary care”, “preventable and originated in primary care”, or “not preventable, standard treatment” [8]. 

Severity was coded as mild (e.g., nausea, rash), moderate (e.g., ongoing poor disease control), severe (e.g., morphine overdose), or death [5].

The records review was conducted by actively practicing general practitioners (family physicians) to enable interpretation of general practice clinical records and improve the identification of patient harms. All records were electronic and used standard practice management software.

### 3.1. Statistical Analysis

Data are presented as summary statistics, namely patient demographic details (age, sex, ethnicity, and socioeconomic deprivation), clinical information (number of consultations and number of unique medications prescribed during the study period), and general practice characteristics (practice size and location). 

Medication-related harms arising from general practice prescribing were examined by ATC classification [9]. The weighting was adjusted for the relative likelihood of each patient’s data being included in the study (i.e., the probability of each practice being selected per strata, and each patient being selected per practice). Weighting was applied to extrapolate the results to the New Zealand population. 

Incidence rates were calculated as the number of events divided by the total number of person–years of follow-up (for example, 3 × 2300 years, 3 years per person). Associations between harm, patient demographics, clinical information, and practice characteristics were investigated using logistic regression with robust standard error. The final model was replicated that was used in the analysis of the full study [10]. Age was treated as a continuous variable, and ethnicity was divided into “Māori” and “non-Māori” for the logistic regression modelling, due to small patient numbers in some groups. The “number of prescriptions” variable was excluded from the multivariable model due to collinearity between that variable and the “number of GP consultations” variable. The estimates were adjusted using appropriate sampling weights.

The statistical analysis used Stata version 17.0 (College Station, TX, USA). The Stata ‘svy’ package was used for applying the sample weights. Data were missing for ethnicity (*n* = 25, 1.1%) and deprivation (*n* = 9, 0.4%). There was no imputation of missing data. Complete data analyses were carried out on 2059 patients.

### 3.2. Ethics

Consent was obtained from the general practices, not from individual patients. All data were de-identified prior to coding and analysis. This research was approved by the University of Otago human ethics committee (HD14/32). The Ngāi Tahu Research Consultation Committee endorsed this research. 

## 4. Results

### 4.1. Demographic and Clinical Results (Table 1)

Patient age ranged from 0 to 20 (median 11, IQR 6–15) years old. The largest age group were aged 2 to <12 years old (1222/2300, 53.1%). There was an even distribution by sex (female 1154, 50.2%). Most participants were NZ European (1539, 67.7%), or Māori (511, 22.5%). Deprivation quintile scores were available for 2081 patients; nearly half were from the least-deprived groups (Deprivation Category 1: 518 (24.9%); Category 2: 473 (22.7%)). 

The number of GP consultations ranged from 0 to 71 over the 3-year period (median 5, IQR 2–10, consultations). The number of unique medications prescribed in general practice over the 3-year period per patient ranged from 0 to 53 (median number of medications 3, IQR 1–6). The number of consultations and medications have been grouped to aid interpretation.

**Table 1 ijerph-20-06378-t001:** Demographic and clinical details for the paediatric patients in SHARP.

		Study Patients N = 2300 (100%)	Patients with Harm N = 141/2300 (6.3%)
Age Group	<2 years	29 (1.3)	1 (3.5)
	2 to <12 years	1222 (53.1)	80 (6.6)
	12 to <18 years	730 (31.7)	36 (4.9)
	18 to ≤20 years	319 (13.9)	24 (7.5)
Sex	Female	1154 (50.2)	77 (6.7)
	Male	1146 (49.8)	64 (5.6)
Ethnicity Missing = 25	NZ European	1539 (67.7)	103 (6.7)
	Māori	511 (22.5)	27 (5.3)
	Pasifika	102 (4.5)	4 (3.9)
	Other	123 (5.4)	4 (3.3)
Deprivation NZDep13 1 = least deprived Missing = 9	1	518 (24.9)	28 (5.4)
	2	473 (22.7)	35 (7.4)
	3	394 (18.9)	20 (5.1)
	4	335 (16.1)	25 (7.5)
	5	361 (17.4)	24 (6.7)
No. GP Consults	0–3	872 (37.9)	6 (0.7)
	4–12	995 (43.3)	56 (5.6)
	13+	433 (18.8)	79 (18.2)
No. GP Rx	0–4	1511 (65.7)	37 (2.5)
	5–9	592 (25.7)	61 (10.3)
	10+	197 (8.6)	43 (21.8)
GP Location	Urban	1116 (48.5)	69 (6.2)
	Rural	1184 (51.5)	72 (6.8)
GP Practice size	Large	820 (35.7)	71 (8.7)
	Med	732 (31.8)	43 (5.9)
	Small	748 (32.5)	27 (3.6)

There was an even distribution of patients enrolled in urban and rural general practices (urban 1116 (48.5%); rural 1184 (51.5%)), and roughly one third of patients were enrolled in each of the large, medium-sized, and small practices (large 820 (35.7%); medium 732 (31.8%); small 748 (32.5%)).

### 4.2. Patient Harms 

There were 193 harms recorded for 141 children and adolescents during the 3-year study period. Among the patients, harms were reported for one patient (3.5%) aged <2 years, 80 patients (6.6%) aged 2 to <12 years, 36 patients (4.9%) aged 12 to <18 years, and 24 patients (7.5%) aged 18 to ≤20 years (Table 2). 

There were 103 (4.5%) patients with one harm, 50 (2.2%) with two harms, 36 (1.6%) with three harms, and four (0.2%) with four harms (Table 3). There were three harms resulting in hospitalization, one of which was medication-related, and there were no deaths.

The annualised rates of harm, weighted to the NZ population, were 36/1000 for the child and adolescent population for all harms, 20/1000 for medication-related harm, and 2/1000 for severe medication-related harms. The rate of hospitalisation secondary to harm was 0.4/1000 for the child and adolescent population. The incidence rate of potentially preventable medication-related harm was 15.6 harms per 1000 patient–years. No children or adolescents died during the study period. 

### 4.3. Medication-Related Harms

For medication-related harms, the drug groups most frequently involved were anti-infectives with 54 (51.9%), genitourinary with 16 (15.4%), dermatological drugs with 13 (12.5%), and the nervous system with 10 (9.6%) (Table 3). In the younger age group, anti-infectives were the most frequent medication causing harm and in the adolescent age groups genitourinary and dermatologicals were the most frequent. These reflect the most common presenting conditions to primary care in these age groups. In addition, there were a further 37 (19.2%) harms that were attributed to medications that were not prescribed in general practice.

### 4.4. Logistic Regression Modelling

Patients ≤20 years were much less likely to experience harm compared to adults in the SHARP study (OR 0.30, 95% CI 0.23, 0.39, *p* < 0.001). Logistic regression modelling was undertaken to determine the association of demographic and clinical variables with child and adolescent harm (Table 4). Increasing numbers of consultations was strongly associated with increased odds of experiencing harm. Compared to patients with 0–3 consultations over the 3-year study period, patients with 4–12 consultations had an odds ratio for experiencing harm of 11.01 (95% CI 2.61, 46.41; *p* = 0.001), and those with 13 or more consultations had an odds ratio of 45.69 (95% CI 11.11, 187.92; *p* < 0.001). 

There may be a complex relationship between socioeconomic deprivation and the odds of experiencing harm, with patients who were in socioeconomic quintile 4 (relatively more deprived) being more likely to experience harm that the other socioeconomic quintiles. Being enrolled in a small general practice was associated with reduced odds of experiencing harm. Age, sex, ethnicity, and location of the general practice were not associated with the risk of experiencing harm. 

## 5. Discussion

The rates of harm in children were lower than in a corresponding adult population, harms were related to the most common treatments used in each age group, and harms increased with the number of consultations. The present analysis was of the child and adolescent subset of the SHARP study. Overall, the risk of harm was 123 harms/1000 patient–years in the full SHARP study analysis, compared with 36/1000 in the child and adolescent population in the present analysis. The results from the full SHARP study, including children and adults, demonstrated an increased risk of harm in patients over the age of 70, with an OR (95% CI) of 3.23 (2.37 to 4.41) [11]. However, that analysis did not examine the very young age groups, which were pooled into the 0 to 49 years (reference) age group. Hence, the previous analysis examined the risk for the very old, but did not examine the risk for the very young. The results of the present study confirm that the population in general practice at greatest risk is the very old, rather than the very young. 

The finding of low rates of treatment-related harm in the paediatric primary care population is consistent with prior data [4]. However, previously this has been interpreted as under-reporting compared to the adult population, whereas in the present study, using record reviews, the same methodology was used to compare the paediatric and adult populations. Spontaneous reporting, as used in previous research, might under-report harms, but this is unlikely when using the methodology in the present study. Consistent with previous reports, the most commonly reported types of incidents were medication incidents [3]. 

Also, there were few reports of serious harm or hospitalisation, and no deaths of children. This contrasts with the rates in the adult population, from the SHARP study, where the estimated incidence of hospitalisation secondary to harm was 2.3 per 1000 patient–years and the estimated incidence of death from harm was 0.5 per 1000 patient–years, with approximately half of the deaths considered to be preventable. Hence, the severity of harms appears to be greater in the adult population. 

Despite the rates and severity of harm being lower in the paediatric population, any harm is significant. This is because of issues around consent and vulnerability. Children rely on carers and health practitioners to make decisions for them, and those decisions should incorporate the risk of harm. Harm may impact on growth and development, and may have lifelong consequences. Harm may discourage children and adolescents from seeking medical care, and from adhering to treatment regimens.

The lower rate of harm in the paediatric and adolescent population may indicate the primary care population is different to the secondary and tertiary care populations. Comparisons of rates of events with tertiary care data is complicated by the different denominators used, e.g., patient–years exposure compared to rates per admission or patient days of admission. The rates of ADEs reported in a New Zealand inpatient paediatric population of 2.1 per 100 prescription episodes, 12.9 per 100 admissions, and 22.1 per 1000 patient days, are difficult to compare with the rate for harm reported in the present study of 36/1000 patient–years [2]. Complaints with regard to healthcare in New Zealand are dealt with by the Health and Disability Commissioner and provide an indicator of healthcare-related harm. In the 6-month period from July to December 2021, 39 (7%) of the complaints from District Health Boards were for the age group 0 to 17 years old (HDC) [12]. This might also indicate lower risks for children and adolescents even within the inpatient population. The estimated risks for an inpatient population are also complicated by the risk for that patient varying with the severity, and time course, of their illness. In addition, children and adolescents in a hospitalised population may have more complex medical conditions and more treatments, and therefore have a greater risk of treatment-related harm than those seen in general practice.

The rates for medication-related harm, in a general population using the same methodology, were lower in children. Leitch 2021 reported the incidence rate of medication-related harm in a general population and in the same setting were 73.9 harms per 1000 patient–years, approximately double the rate in children reported in the present study [10]. In that study, when expressed in terms of the odds ratios in comparison with a reference population of 15 to 59 years, the OR (95%) CI for medication-related harm was 0.75 (0.42 to 1.33) for 0 to 4-year olds, 0.58 (0.31 to 1.10) for 5 to 14-year olds, 1.98 (1.50 to 2.61) for 60 to 74-year olds, and 3.08 (2.15 to 4.41) for patients aged >75 years. This indicates that older general practice patients are at increased risk from medication-related harm, rather than the very young. 

Harm appears to be associated with the most common treatments used in each age group. In the younger age group anti-infectives were the most frequent medication attributed to harm, while genitourinary and dermatological medications caused most harm in the adolescent age groups. Febrile illness is the most common presenting problem in young children and anti-infectives are the most commonly prescribed medications [13,14]. Given the inherent risks associated with antibiotics, primarily gastrointestinal upset and hypersensitivity, their correlation with harm in the younger age groups is not unexpected. In adolescents, although the most commonly prescribed group of drugs is anti-asthma medications, control of the reproductive cycle in girls, and acne treatments in both sexes are common reasons for initial and repeat presentations [15]. Hormonal and anti-acne treatments have known adverse effect profiles, therefore the association of genitourinary and dermatologic drug classes with harm in this age group is also not unexpected. 

The medications commonly associated with harm in the adult general practice population differ to the paediatric population [10]. In adults, these were cardiovascular medications, antineoplastic and immunomodulatory agents, and medication relating to blood and blood forming agents. Hence, the medications associated with harm relate to the predominant medical conditions in each age group, and interventions to prevent harm may differ between age groups.

Vaccinations would be expected to have a stronger representation in the present study, but the adverse events from these tend to be minor and may not have resulted in attending a general practice. In New Zealand, childhood vaccinations are usually administered by nurses. The adverse events from vaccinations are also primarily dealt with by nurses. The New Zealand Centre for Adverse Reaction Monitoring received a total of 3581 reports of suspected adverse reactions in 2021, and of these 1470 (41%) were for vaccines [16]. Nurses were the professional group that reported the most suspected adverse reactions and, as these data do not include suspected adverse reactions to COVID-19 vaccines, children can be presumed to be the groups most affected. The majority of these adverse reactions can be treated with simple analgesia, but the more serious may have resulted in presentation to the emergency department. Previous research into paediatric vaccinations reported that failure to vaccinate, and the mistiming of vaccinations, resulted in more serious harm than the vaccination itself [17].

The risk of harm increasing with the number of consultations may have several interpretations. Patients with chronic conditions will have more consultations, and more complex and potentially hazardous treatment regimens. Where initial treatment has failed, subsequent treatments may be used that have a lower benefit–risk balance. There is a paucity of research investigating these potential factors. Hence, more research is required to determine the reasons why patients who have more consultations have higher rates of treatment-related harm.

The present study is limited by the age of the data, the lack of detail around the circumstances of the events and the method for urban–rural classification that was used. In New Zealand, the care of the newborn in primary care is the responsibility of midwives up to the age of 6 weeks, so neonates will be under-represented in this dataset. The age of the data is unlikely to bias the results because primary care practice has not changed substantially since the study was performed. The retrospective study and use of general practice records as the primary data source means that issues such as extemporaneous formulations, off-label use, and unlicensed medicine use were unable to be assessed [4]. There is also no information as to which is the most significant risk point in the paediatric medication pathway: prescribing, transcribing, dispensing/labelling, administration or monitoring [4]. A prospective study would have been able to examine these issues in more detail. The use of the older Statistics New Zealand classification may underestimate the burden of disease in the rural population, but this may not impact on the risk of treatment-related harm [18]. 

The main strength of the study is the thorough review of the medical records, which contributes to completeness of the data. The present study also enables a direct comparison with the adult population, using the same methodology and in the same setting. 

Potential interventions to decrease treatment-related harm in primary care include providing enhanced drug information (such as the New Zealand Formulary) for prescribers, clinical decision support that takes into account the age and size of the patient, the use of dosing calculators (such as the Paediatric Analgesia Wheel [19]), and electronic prescribing combined with clinical decision support customised for children [4]. The New Zealand Formulary, just like the British National Formulary, has a separate edition for children [20]. 

A process for developing such interventions has been recommended by Sutcliffe (2014) as “customisation for use with child patients, engaging with a range of stakeholders during development, fostering a high level of familiarity with the system prior to use, ensuring adequate IT systems and compatibility with existing hospital systems and infrastructure, careful planning and ongoing iterative development post-implementation” [4]. This same process could be adapted for primary care.

## 6. Conclusions

In a New Zealand primary care population, treatment-related harm in children was less common than in a corresponding adult population. Medication-related harm was the most common type of harm and was related to the most common treatments used in each age group. The risk of harm increased with the number of consultations. Avoiding medication-related harm requires a multifaceted systems-level approach developed with stakeholders.

## Figures and Tables

**Table 2 ijerph-20-06378-t002:** Harms experienced by paediatric patients in SHARP.

		Harms N = 193 (100%)
Harm type	Medicines	141 (73.1)
	Surgery	18 (9.3)
	Access	12 (6.2)
	Communication	9 (4.7)
	Other	9 (4.7)
	Investigations	4 (2.1)
Number of harms per patient	1	103 (53.4)
	2	50 (25.9)
	3	36 (18.7)
	4	4 (2.1)
Preventability	Preventable	32 (16.6)
	Not preventable	161 (83.4)
Severity	Mild	150 (77.7)
	Moderate	37 (19.2)
	Severe	6 (3.1)
	Death	0
Hospitalised	Not hospitalised	190 (98.4)
	Hospitalised	3 (1.5)

**Table 3 ijerph-20-06378-t003:** Medication-related harm by ATC category, frequency, and age group associated with prescribing in general practice (*n* = 104).

	Age Group			
ATC Category *	<2 Years	2 to <12 Years	12 to <18 Years	18 to ≤20 Years
J. Anti-infectives for systemic use	0	43	9	2
G. Genitourinary and sex hormones	0	0	5	11
D. Dermatologicals	0	1	6	6
N. Nervous system	0	5	4	1
A. Alimentary tract and metabolism	0	0	0	2
B. Blood and blood forming organs	0	0	1	1
P. Antiparasitic products	0	1	1	0
R. Respiratory system	1	0	1	0
V. Various	0	0	1	1
M. Musculoskeletal system	0	1	0	0
TOTAL	1	51	28	24

* No harms were recorded for the following ATC classifications: C. Cardiovascular system, H. Systemic hormonal preparations, L. Antineoplastic and immunomodulating agents, S. Sensory organs.

**Table 4 ijerph-20-06378-t004:** Risk factors for paediatric patients experiencing harm.

		Adjusted and Weighted OR (95% CI)	*p* Value
Age	0 ≤ 20 years	1.02 (0.97, 1.07)	0.524
Sex	Male	Ref	
	Female	1.45 (0.85, 2.49)	0.175
Ethnicity Missing = 25	Non-Māori	Ref	
	Māori	0.95 (0.46, 1.94)	0.888
Deprivation NZDep13 1 = least deprived Missing = 9	1	Ref	
	2	1.28 (0.61, 2.68)	0.512
	3	1.06 (0.44, 2.54)	0.891
	4	2.39 (1.09, 5.22)	0.029
	5	1.91 (0.82, 4.46)	0.133
No. GP Consults	0–3	Ref	
	4–12	11.01 (2.61, 46.41)	0.001
	13+	45.69 (11.11, 187.92)	<0.001
GP Location	Urban	Ref	
	Rural	0.85 (0.55, 1.31)	0.466
GP Practice size	Large	Ref	
	Med	0.69 (0.40, 1.17)	0.165
	Small	0.54 (0.30, 0.97)	0.039

## Data Availability

Due to confidentiality agreements, the data are not publicly available.

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
