# Peer review of "Treatment Harms in Paediatric Primary Care"

_ijerph, 2023, doi:10.3390/ijerph20146378_

Round 1

Reviewer 1 Report

Dear Authors,

Thank you for this interesting paper regarding Harm in the pediatric and adolescent population.

The paper in general is well written.

But I have the following comments:

·       Significance presented by p-value should also be determined for Harm experienced by pediatric patients not only for the association of demographic and clinical variables.

·        Line 210-212 “The results from the full SHARP study, ………. including children and adults, demonstrated an increased risk of harm in patients over the age of 70, with an OR (95% CI) of 3.23 (2.37 to 4.41) [11]"

So, in the main SHARP study in reference 11: “Reviewers identified 2972 harms affecting 1505 patients aged 0–102 years.” Having the same conclusion in both studies; What is the difference between your study and the main study? Kindly clarify.

Author Response

Reviewer 1:

Dear Authors,

Thank you for this interesting paper regarding Harm in the pediatric and adolescent population.

The paper in general is well written.

But I have the following comments:

  • Significance presented by p-value should also be determined for Harm experienced by pediatric patients not only for the association of demographic and clinical variables.

Response:

In lines 186 to 187 we tested the risk of harm in patients <20 years compared to adults in the as stated in the text: “Patients ≤20 years were much less likely to experience harm compared to adults in the SHARP study (OR 0.30, 95%CI 0.23, 0.39, p <0.001).” 

Reviewer 1: 

  • Line 210-212 “The results from the full SHARP study, ………. including children and adults, demonstrated an increased risk of harm in patients over the age of 70, with an OR (95% CI) of 3.23 (2.37 to 4.41) [11]"

So, in the main SHARP study in reference 11: “Reviewers identified 2972 harms affecting 1505 patients aged 0–102 years.” Having the same conclusion in both studies; What is the difference between your study and the main study? Kindly clarify.

Response:

Lines 212 to 215 state “However, that analysis did not examine the very young age groups, which were pooled into the 0 to 49 years (reference) age group.  The results of the present study confirm that the population in general practice at greatest risk is the very old, rather than the very young.”

We have added the following sentence into this section to further clarify: “Hence, the previous analysis examined the risk for the very old but did not examine the risk for the very young.”

Reviewer 2 Report

This current study is a sub-group analysis of the SHARP study. The age group was ≤ 20 years age.

Major comments:

11.       Line 99-101 “The records review was conducted by actively practicing general practitioners 99 (family physicians) to enable interpretation of general practice clinical records and improve the identification of patient harms. All records were electronic and used standard practice management software.”

è We need to read the previous SHARP studies to have detailed information about methodology. I suggest that this above paragraph would be more detailed about how harms were identified and checked for causal relationship with health system. Which method was used to assess causal relationship?

22.       It would be more relevant to give more detailed about MRH (for dosing, formulation, off label use or adverse drug reaction…?)

33.       Why the number of drugs prescribed per child, off-label use was not tested as risk factors.

Minor points:

Line 94: Mc Kay et al 2103 à to be corrected as 2013

Author Response

Reviewer 2:

This current study is a sub-group analysis of the SHARP study. The age group was ≤ 20 years age.

Major comments:

  1. Line 99-101 “The records review was conducted by actively practicing general practitioners 99 (family physicians) to enable interpretation of general practice clinical records and improve the identification of patient harms. All records were electronic and used standard practice management software.”

è We need to read the previous SHARP studies to have detailed information about methodology. I suggest that this above paragraph would be more detailed about how harms were identified and checked for causal relationship with health system. Which method was used to assess causal relationship?

Response:

Harms were identified as “physical, emotional, or financial negative consequences to patients directly arising from health care, beyond the usual consequences of care and not attributable to patients’ health condition.”  The GP reviewers identified these from a longitudinal review of the patients’ health records.  Hence, the alignment of cause and effect was dependent on the experience of the GP reviewers.  A formal causality assessment was not performed and is not reported in the manuscript.  The operational definition of patient harm was derived from work undertaken by the Australian Council for Safety and Quality in Health Care.

  The published protocol describes the methodology in detail and is available open-access.  The methods were reviewed by the Health Research Council of New Zealand and the study was successful in being funded by the HRC following an extremely competitive process.

The authors accept that causality assessment is important in the assessment of adverse drug reactions, but in the field of research into patient harm, causality assessment has not been considered to be an imperative.

In order to clarify the identification of harm, the following has been inserted into paragraph 3 of methods: “Harm was identified by the investigators from a longitudinal review of the selected individual patient records.”

Reviewer 2:

  1. It would be more relevant to give more detailed about MRH (for dosing, formulation, off label use or adverse drug reaction…?)

Response:

For medication related harm (MRH), to examine dosing would have required a different dataset, which would have looked at the appropriateness of the dose (per weight and New Zealand Formulary dosing).  This would have required a different set of reviewers (pharmacists as opposed to general practitioners).  To examine formulation would have required dispensing data, because the formulation may not have been specified on the prescription.  This would have required a different dataset, i.e., community pharmacy data.  Off-label use is often a subjective assessment, as it can depend in the indication as well as the age group.  Hence, this would have required reviewers who were paediatricians.  However, the scope of the study was to examine the context of general practice, and therefore general practice reviewers were considered to be suitable.

Reviewer 2:

  1. Why the number of drugs prescribed per child, off-label use was not tested as risk factors.

Response:

Off-label use was not included as a variable in the original SHARP study data.  Hence, examining this variable would have required a re-analysis of the original data, and re-coding.  This would have required considerable resource.  In New Zealand, off-label use is supported by the New Zealand Formulary for Children which includes dosing advice and advice on the appropriateness of off-label indications.  The vast majority of off-label use in New Zealand is supported by data for dosing, safety and efficacy and this information is available to all New Zealand residents but restricted from overseas views by blocking non-New Zealand IP addresses.  Hence, in general practice usage, and given the drugs actually prescribed to the cohort, off-label use would be unlikely to feature as a risk factor.  Therefore, we could not justify the extra resource to investigate this variable.  However, we did acknowledge this in the limitations section in the sentence: “The retrospective study and use of general practice records as primary data source means that issues such as extemporaneous formulations, off-label use and unlicensed medicine use were unable to be assessed”.  Number of drugs prescribed was examined, but was highly correlated with number of consultations, which was a more significant variable.  Following statistical advice, highly correlated variables were not included in the final model.

Reviewer 2:

Minor points:

Line 94: Mc Kay et al 2103 à to be corrected as 2013

Response:

This has been corrected.

Reviewer 3 Report

Dear Authors

The study entitled "Treatment Harms in Paediatric Primary Care" was conducted with the objective to describe the epidemiology in children of harms detect able from general practice records, and to identify risk factors.

The manuscript was written well.

I have some questions and suggestions:

- Write your aim/objective in the abstract.

- There were 210,559 patients enrolled. On what basis, you selected only 9076 patient records?

- From 9076, you included 2300 patients. Was participants of ≤20 years age your only inclusion criteria? Explain the inclusion and exclusion criteria in the methodology.

- Line 141 - The number of unique medications prescribed in general practice over the three-year period per patient ranged from 0-53 (median number of medications 142 3, IQR 1-6). For 1 patient, what is the reason fro prescribing 53 medications? Is it a typo error that instead of 5 you wrote as 53?

- What was the treatment given for treating the harms?

Author Response

Reviewer 3:

Dear Authors

The study entitled "Treatment Harms in Paediatric Primary Care" was conducted with the objective to describe the epidemiology in children of harms detect able from general practice records, and to identify risk factors.

The manuscript was written well.

I have some questions and suggestions:

  • Write your aim/objective in the abstract.

Response:

The first sentence of the abstract has been changed from background to aim/objective with the sentence: “The aim of the study was to describe the epidemiology in children of harms detectable from general practice records, and to identify risk factors.”

Reviewer 3:

  • There were 210,559 patients enrolled. On what basis, you selected only 9076 patient records?

Response:

The sampling was from randomly selected general practices consenting to participate in the study, from which was drawn a random selection of patients enrolled at the midpoint of the study period, July 1, 2012.  The sample was balanced by practice size and location.

This section of the methods has been expanded to: “Patients were selected at random from those enrolled with the general practices chosen for inclusion in the study, and the sampling was balanced by practice size and location.”

Reviewer 3:

  • From 9076, you included 2300 patients. Was participants of ≤20 years age your only inclusion criteria? Explain the inclusion and exclusion criteria in the methodology.

Response:

The inclusion criteria have been expanded to: “The age group for the purposes of this study was from birth to ≤20 years age, consisting of 2300 SHARP patients.  Males and females were included and there were no exclusion criteria.”

Reviewer 3:

- Line 141 - The number of unique medications prescribed in general practice over the three-year period per patient ranged from 0-53 (median number of medications 142 3, IQR 1-6). For 1 patient, what is the reason for prescribing 53 medications? Is it a typo error that instead of 5 you wrote as 53?

Response:

The study was conducted over a three-year period, during which time patients attended their general practice multiple times and received many prescriptions.  A single medication may have been re-prescribed on multiple occasions, (e.g., salbutamol),  But this would have only been counted as one unique medication. The patient who received 53 unique medications was an outlier, as indicated by the median and interquartile range for the cohort.

Reviewer 3:

- What was the treatment given for treating the harms?

Response:

This data was not collected as it was beyond the scope of the study.  The aim/objective of the SHARP study was to describe the epidemiology in children of harms detectable from general practice records, and to identify risk factors.  This did not include identifying treatments.